# A new model for intra- and inter-institutional soil data sharing

José Padarian and Alex B. McBratney

Sydney Institute of Agriculture & School of Life and Environmental Sciences, The University of Sydney, New South Wales, Australia

**Correspondence:** José Padarian (jose.padarian@sydney.edu.au)

**Abstract.** Data sharing and collaboration are critical to solving large scale problems. The prevailing soil data-sharing model is based on different groups sending their data to a lead party. This model is of a centralised nature and, consequently, results in the participants ceding control and governance over their data to the lead party. Here we explore the use of a distributed ledger (blockchain) to solve the aforementioned issues. We explain what a blockchain is and some of its characteristics to then describe some features of a blockchain that makes it an interesting candidate for an inter-institutional database. Finally, we describe the potential use case of developing a global soil spectral library with multiple, independent international institutions constituting the network.

## 1 Introduction

Soil is a key component of ecosystems and the need for soil information to monitor its condition is increasing. A large amount of soil data has been collected in the last century, with a special increment during the 70s-80s, and many organisations are performing the exhaustive task of "rescuing" and organising that data in more accessible formats (Arrouays et al., 2017). Additionally, in many countries, a large amount of new soil data is being generated partially aided by the advancements in methods such as soil spectroscopy, which makes the acquisition of soil data faster and cheaper compared with traditional wet chemistry methods (Brown et al., 2006; McBratney et al., 2006).

Most collected soil data is useful to solve problems locally but it is too fragmented to tackle more general issues. This applies at various levels of granularity including different teams within an institution, a single institution in different regional locations, and multiple institutions either within a country or internationally. In these cases, collaboration and data sharing becomes paramount. The soil community recognises this collaboration need and has responded by creating different data-sharing initiatives. For instance, Rossel et al. (2016) compiled a global soil spectral library for soil mapping, modelling and monitoring with datasets from 92 countries (mainly data from United States, Australia and Europe). Another global spectral library has been promoted by FAO's Global Soil Partnership via the Global Soil Laboratory Network (GLOSOLAN). FAO also promoted different initiatives to establish collaboration networks to share soil profile information, including between Latin American countries (SISLAC) or a Global Soil Information System (GLOSIS). All these initiatives are designed as

centralised systems were, in order to collaborate, different parties (either individuals or organisations) should send their data to the lead organisation (Fig. 2a). A centralised information system has a series of disadvantages that we explore in this work. Two of the most important disadvantages of using a centralised network, especially in the context of a collaborative network of multiple independent parties, are that the control and data governance (norms, principles and rules) are completely ceded to the initiating party. Usually, these aspects could be defined in a data-sharing agreement prior to the establishment of the collaboration but, in practice, there are no controls to avoid unilateral decisions.

A potential solution for the data control and governance issues derived from the implementation of a centralised data-sharing system in the use of distributed a ledger or blockchain. The aim of this paper is to delineate the requirements for a functional, decentralised, inter-institutional database (IIDB) to share soil information on a distributed ledger or blockchain. We mainly focus on the technical considerations of data sharing instead of its social, political and organisational aspect, keeping in mind that the latter are important for any data-sharing system, decentralised or not. First, we introduce some terms that will be used throughout this paper to then explain what a blockchain is and some of its characteristics. Second, we describe some features of a blockchain that makes it an interesting candidate for an IIDB. Finally, we present a use case of collaborative effort that could be a good fit for using the proposed model.

## 2   Blockchain

Before defining what a blockchain is, we introduce a list of definitions that are used throughout this paper:

**Point of failure:** A potential risk caused by a poor system design were a single fault at that point can affect the correct functioning of the system.

**Hash:** Alpha-numeric string generated by mapping the data of an arbitrary size onto data of a fixed size (Dworkin, 2015). When the original data is unknown, it is very difficult to reconstruct it from the hash value, which makes it a good candidate to ensure the integrity of a transaction.

In simple terms, a blockchain is a linked sequence of records of the transactions of digital assets (Fig. 1). These transactions can be of different types, including data creation (adding new data to the blockchain) or transfer (transferring the ownership of the data to another party, or to the same owner to edit data). The best known assets are crypto-currencies (e.g.: bitcoin), but in practice can be anything that can be represented by data. Each new transaction is cryptographically signed using the party's private key which is verified against the public key (included in the asset). The transaction also includes a hash that is generated using its public-private key-pair and the hash of the previous block. Any attempts to modify a block that has already been incorporated into the blockchain would change the signature and the hash of that transaction, which can be detected.

It is worth detailing what a key-pair is and how it operates in the context of signing transactions. In asymmetric cryptography, two keys are used — private and public keys (Kumar et al., 2011). The private key is used to generate a signature based on the data included in the transaction and the public key is used to verify the signature. As the names suggest, the private key is only known by the signing party, and the public key is available to everyone to verify the signature. Generally, the signature

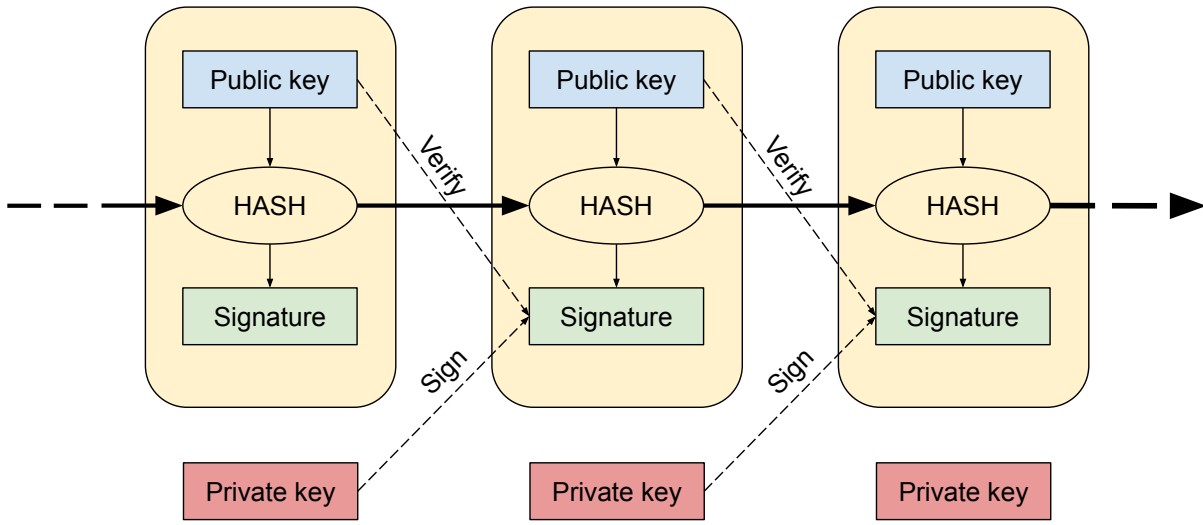

**Figure 1.** Diagram of three consecutive blocks (two transactions) within a blockchain.

and the public-private key-pair are long sequences of alpha-numeric characters which are algorithmically generated. There are many implementations of these algorithms but, in general, the algorithm to generate the signature $s$ can be rationalised as a function $s = f(h, k_{priv})$, where $h$ is the hash of the data to sign and $k_{priv}$ is the private key. In order to verify the signature, the verifier should compute the hash $h'$ using the signature $s$ and the public key $k_{pub}$ such as $h' = g(s, k_{pub})$ and also independently compute the hash of the data $h$ using the same hashing algorithm (publicly known). If $h = h'$, the signature is valid.

By design, a blockchain usually operates within a network of interconnected nodes (Fig. 2b). Each node keeps a copy of the chain (public ledger) and acts as validator, assuring the validity of new transactions. After enough nodes have reached consensus about the validity of the transaction, the new data-block is appended to the chain.

Blockchain technology is a diverse ecosystem with many implementations that differ in their characteristics and efficiency. For instance, popular implementations such as Bitcoin, Ethereum, Litecoin and Monero are computation-intensive and require large energy input due to their consensus algorithm (proof-of-work), consuming more energy than mineral mining (copper, gold, platinum and rare earth oxides) to produce an equivalent market value (Krause and Tolaymat, 2018). Of course, other consensus algorithms (e.g. proof-of-stake) do not require intensive computations. Given the diversity of implementations, it is completely possible to design a system that is secure, reliable and efficient to serve as a soil data-sharing platform.

## 3 A soil data-sharing platform based on blockchain

Besides providing a solution to the aforementioned problems, namely centralised data control and governance, a blockchain has other characteristics that makes it an interesting candidate for a IIDB. Some of these solutions and characteristics are described in this section.

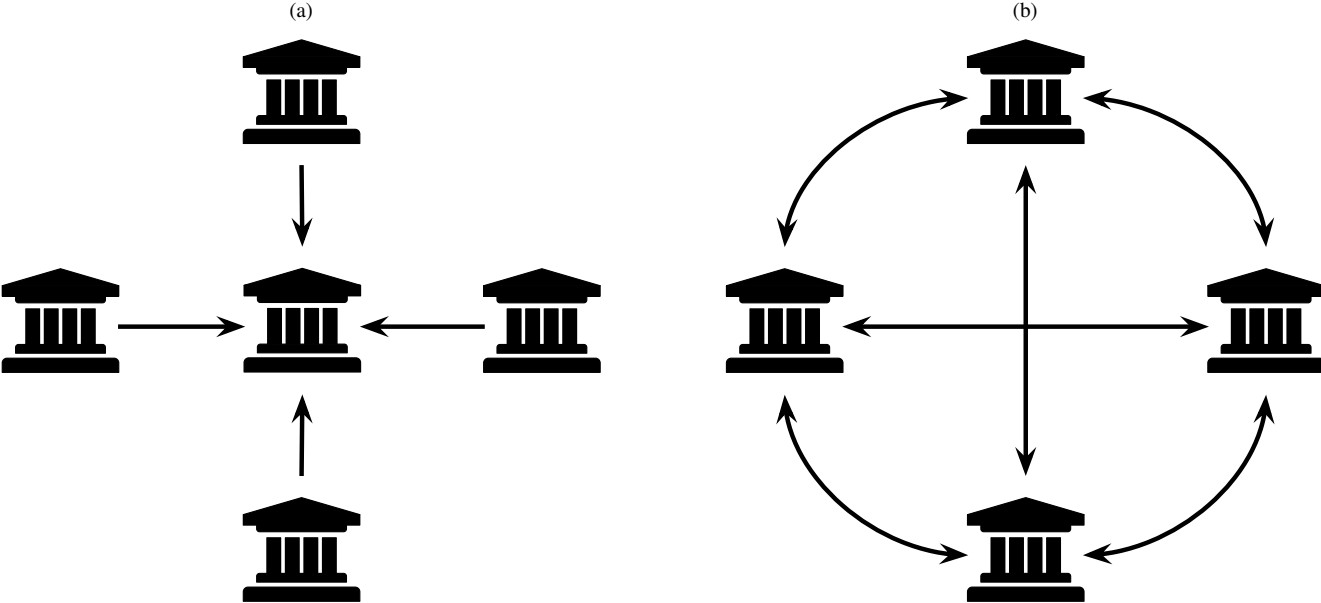

**Figure 2.** Data flows in two different soil information systems infrastructures (a) centralised (b) decentralised.

## 3.1 Decentralisation

As mentioned before, the main characteristic of a blockchain is the decentralised nature of the system. Each node of the network keeps a copy of the blockchain, which is synchronised after every new transaction (creation or transfer). Assuming that each node of the network is controlled by a different party, there is no centralised data storage, hence no single point of failure or control. Normally, in a well-designed, diverse network, a significant number of the nodes can be compromised without affecting its integrity.

Because all the nodes have a copy of the blockchain and act as validators, malicious modifications to the data are very difficult (see immutability section). The only possible way of tampering the data is if most of the nodes are colluded, which can be avoided by ensuring a diverse network.

For intra-institutional data sharing, a blockchain system can also be implemented to replace a traditional, permissioned database. The advantages are similar to the inter-institutional case, including each team leader having "ownership" of their data, data traceability, data access logging and potentially preventing unauthorised access, and preventing malicious modifications or deletions. Data is one of the most valuable assets of any company and adding this extra layer of security to ensure its integrity should be a priority, and even mandatory for publicly funded institutions.

## 3.2 Data Governance

Data governance defines the norms, principles and rules under which the activities of a consortium should be conducted. It might include important details such as data release and rights to publish with consortium data first, research output rules (e.g. authorship sequence in consortium publications), if the data should be shared with non-consortium members (Singh and Daar, 2009), and the addition of new members. In a data-sharing network, data governance is stipulated on an agreement and any modifications can be agreed between the members. In practice, control and governance over the data are ceded to the central node and the system has no way to prevent that unilateral changes are made.

Using a technology such as blockchain does not replace the initial process of negotiations nor the effort of setting rules but it can help reducing some of the friction points. Many of the clauses included in a data-sharing agreement can be programmatically enforced and, since the network is collectively governed, changed over time via a democratic process. Usually, any node of the network can propose an election process where the rest of the nodes cast a vote transaction, which is also appended to the chain. If the "super-majority" (usually a large proportion such as at least 2/3) of nodes approves the changes, the suggested changes are incorporated into the system.

## 3.3 Data ownership

When a new asset is created, it is cryptographically signed and assigned to one or more users' public-key(s). If the data needs to be transferred (either to make corrections or include new information, or to another user), only the owner are capable of doing so by using the corresponding private keys, even if the whole blockchain data is available at every node. This process is automatically validated by all the nodes by ensuring that the signatures match with the owner(s) public-key(s) before proceeding with the transfer.

Here we refer to data ownership as the link between an user and a digital asset, without any legal implication. Like in any database, decentralised or not, we are assuming that the user has legal rights to upload the data, which should be properly acknowledged, following the rules defined by the consortium. All this information can be included within each asset, permanently linking data and metadata, where any change can be recorded in case of ownership changes. If required, the network can perform basic checks to ensure that the metadata is included or even just provide access to encrypted data to authorised users.

## 3.4 Immutability

Since the blocks of the chain are linked (Fig. 1), in practice, it is not possible to remove or edit a transaction. When a party needs to change the content of some asset that they own, either to correct some error or add new information to an asset, a *transfer* transaction needs to be performed, transferring the asset to itself. When the transaction is approved by the network, the new version is appended to the blockchain. This design feature produces data redundancy but also makes possible to keep the history of every asset (versioning), which is key for auditability. Thanks to this immutability, the parties within the network can always trust that the data has not been tampered.

Similar to the data ownership case, here we assume that the asset contains data that is legitimate and error-free. In any system, decentralised or not, it is difficult to control what happen to the data before its ingestion into the system. Although it could be possible to implement pre-ingestion solutions, probably it would always be possible to "cheat the system". It is important to consider that there are implicit incentives for the parties to provide legitimate data, especially considering the transparency of a decentralised system (ownership and immutability), such as maintaining their credibility.

## 4  Potential use case: global soil spectral library

Although a blockchain data-sharing model has applications at many levels of granularity (inter- and intra-institutional, and international), we would like to focus on the use case of creating a multi-party (e.g., multi-institutional, multi-national, global) soil spectral library. Spectral soil data can be compared to the digital fingerprint of a particular soil sample which encodes information about its physical, chemical and biological properties (Grunwald, 2016). In pedometrics, a discipline that applies quantitative methods to study the variation of soils, the use of spectral data in conjunction with statistical or machine learning models to predict soil properties is already broadly implemented (McBratney et al., 2006; Nocita et al., 2015; Padarian et al., 2019b). Nevertheless, the development and application of models derived from spectral data still presents a series of challenges. For instance, models derived from local data, despite showing a good performance, have a limited applicability to other areas since they might lose their validity (Grinand et al., 2008). A potential solution is to develop models trained on data obtained from larger extents, which can be then "localised", taking advantage of the global knowledge to make predictions at the local scale (Padarian et al., 2019a). This approach has shown significant improvements of local predictions for multiple soil properties. To take full advantage of these advanced models, and since they are considered as "data-hungry" methods, it is recommended to train them on a large soil spectral library. Of course, collating a large spectral library that spans a large extent is not a trivial task. This is when collaboration and data sharing becomes important. Multiple, independent organisations can join efforts to reach a solution to a problem that it is very difficult to solve independently, which yields institutional (local) benefits greater than what it is possible working in isolation.

After all the efforts from different institutions to collaborate in a common initiative, it is only fair that the data-sharing infrastructure is carefully designed to ensure a democratic access, and control and governance over the data. We believe that, in general, a decentralised system can guard those interest for all parties involved. Particularly in the case of a global soil spectral library, the use of a decentralised database is of critical importance since the resulting database could be used by national reference centres for soil analysis. The level of transparency and security that a distributed ledger offers ensures that the reference data has not been tampered and also, given its decentralised model, will maximise accessibility. In the following sections we explore certain implementation aspects of a decentralised data-sharing system in the context of a global soil spectral library.

## 4.1 Consortium initiation

The potential members of the consortium would have enough analytical capacity to measure the spectral response of soil samples and also to perform laboratory analyses to measure the corresponding physical, chemical and biological soil properties. This includes universities and commercial soil laboratories from different countries.

Each member should have available the computational infrastructure to become a node of the network. The requirements are not prohibitive and include enough capacity to store all the data and internet connection. Each node should generate their public-private key-pair, securely store a copy of the private key and distribute the public key to the rest of the members. To start the network, all the public keys should be known by all the members. Once the network is functional, more nodes can be added with the approval of most of the current members via an election process.

In terms of the network users, it is possible to have multiple users per node (e.g. different researchers from a single University). Ideally, all the users should have their own public-private key-pair to sign their transactions, and their public keys should be known to all the users. This information can also be stored in the blockchain as a public ledger of who can access the data.

As mentioned in Section 3.2, here we do not consider the legal/organisational aspects of creating a consortium, which, arguably, are more complex than the technical challenges. What voting power should each party have (affecting governance)? should it be proportional to the data they provide? how to reference the data? should all the contributors be co-authors of the publications derived from the database? These are some of the question that should be revolved before implementing a technical solution such as a database, decentralised or not.

## 4.2 Providing data

After the network is functional, any member can create new transactions to add data that will be synchronised between all the nodes, ensuring immediate accessibility to the data to all the members. The structure of what constitutes an "asset" should be defined during the consortium initiation period. For instance, the asset could be a single soil sample with its corresponding analytical data (Snippet 1). The system should support the use of numerical and text data to store all the necessary soil properties and metadata. Complex data structures such as soil spectral data can be stored as comma separated numbers or compressed.

The new transaction should be signed with the user's private key and the asset ownership set to a user's public key. This provides a way of authenticating the origin of the data and allows the user, and only that user, to create updated versions of that asset if needed (e.g. when new properties are measured or to correct potential errors). Before a new transaction is appended to the blockchain, a "super-majority" of the voting power must agree on the validity of that transaction. The most basic validation is to assure that the owner(s) are signing the transaction, but in practice it is possible to set any logical rules. This provides the opportunity to give certain groups of users the control over an asset, define minimum number of owners, perform basic data integrity checks (plausible values, names encoding), etc. Of course, as mentioned in Section 3.4, the legitimacy of the data is hard to prove, which should be considered when designing the system.

**Snippet 1.** Example *Sample* asset. The soil property codes are just for illustration purposes.

```
{
    "date": "2019−05−27",
    "user": "University1",
    "top": 0,
    "bottom": 10,
    "oc": 5.2,
    "sand": 10,
    "silt": 20,
    "clay": 70,
    "bd": 1.1,
    "spectra": "EOhM2lSd6j/o19ZP/9nqP+..."   # This spectra is encoded as base64
}
```

## 4.3   Retrieving data

Since every node keeps a copy of the blockchain locally, it is possible to retrieve data from any node from the network, providing extra redundancy and hence assuring accessibility in case of malfunction of some of the nodes. Advanced users can query their local copy of the database directly. A friendlier way of providing access to read the data is via an Application Programming
5   Interface (API) that connects any user with a node. That API can perform tasks such as querying the blockchain to retrieve specific data, provide the history of any asset, and potentially process data using pipelines approved by the consortium.

## 4.4   User interaction

Most of the specific blockchain operations (i.e. signing and verifying transaction) are performed in the background. There is no extra overhead for the users besides keeping their respective private keys safe. A user interface can be build on top of an API
10   so users can access the system as it were a traditional data management system (DMS), with capabilities to query and retrieve the data from the network.

In terms of the type of users with access to the system, any person with access to a node have complete reading access to the blockchain. If public access is required to allow non-consortium members to connect to the database, multiple solutions are available including single or multiple nodes acting as a web server. Using multiple nodes as web servers might reduce
15   latency, specially when the consortium spans different countries (i.e. an external user can connect to the closest node). Again, a platform can be build to ensure the public experience is identical to a normal DMS.

## 5   Summary

The prevailing soil data-sharing model is centralised, with users ceding control and governance over their data to a lead party. We propose the use of a public ledger (blockchain) to create a decentralised soil data-sharing network. This network provides
20   a series of advantages to the participant institutions, including:

- allowing institutions to preserve the ownership and control over their data,

- instant access to the complete database,

- ensures that once the data is appended to the blockchain it cannot be tampered,

- actively participate in governance decisions such as adding new members through elections facilitated by the system.

Ultimately, any consortium data-sharing agreement is based on trust between the participants. By using a blockchain network, the need of trust is removed since rules can be programmatically enforced and the data becomes tamper-resistant. This protects the already existing trust-bond between the consortium members and, potentially, allows the consortium to expand its reach by working with new parties that are not fully-trusted.

For intra-institutional data sharing, a blockchain system can also be implemented to replace a traditional, permissioned database. The advantages include each team leader having "ownership" of their data, data traceability, data access logging and potentially preventing unauthorised access, and preventing malicious modifications or deletions.

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
