# Peer review of "A new model for intra- and inter-institutional soil data sharing"

_SOIL, 2019_

## Referee Comment (RC1) · Dominique Arrouays (Referee) · 10 Oct 2019

This is a nice and timely short paper. Indeed data privacy is a big issue for global science (and expecially for soil for which legal constraints may apply) and the solution proposed here might interest readers of SOIL.

I have only a few minor suggestions of technical corrections, and one for discussion

Page 4 line 17 (ref) the reference is missing

Page 6 Snippet 1; I'm not sure this example is useful if you don't detail more in depth the content. Either skipp it of be more comprehensive.

Page 6 line 11. This election process might be difficult to establish (I mean the rules).

Is it based on individuals, is it proportional to the data you provide as input? to the number of times you provide new data ? This might not be simple and lead to endless discussions. I think you should discuss this. Who establishes the rules? How is the consortium formed. etc. As it stands it seems very simple, but may be it is a bit naive to think that it will work smoothly without a lot of prior negociations?

Cites and references You cite and ref Rossel. it should be Viscarra Rossel (name) and R. (for Raphael) the initial.

---

## Author Comment (AC1) · 14 Oct 2019

Thanks for your comments Dominique.

Regarding your comment:

**Page 6 line 11. This election process might be difficult to establish (I mean the rules).
Is it based on individuals, is it proportional to the data you provide as input? to the number of times you provide new data? This might not be simple and lead to endless discussions. I think you should discuss this. Who establishes the rules? How is the consortium formed. etc. As it stands it seems very simple, but may be it is a bit naive to think that it will work smoothly without a lot of prior**

[Figure]

**negotiations?**

In this paper, and particularly in Section 5.1, we focused on the technical implications of using blockchain instead of the legal/organisational aspects of creating a consortium. From a technical point of view, any of the situations you mentioned can be implemented and changed over time via a democratic process aided by a blockchain-based platform, but such platform cannot replace the initial negotiations nor the effort of setting rules. Hopefully, some of the technical advantages of using a blockchain-based platform (e.g. immediate data access, maintaining control and governance over the data) will reduce the number of points to negotiate. We will add an extra paragraph to clarify that and raise some of the challenges you mentioned.

————————————————————

---

## Short Comment (SC1) · 18 Oct 2019

P1 L 22 to P1 L5

GloSIS is being built in exactly the opposite manner. GLOSIS is envisioned as a federation of soil information systems, which share interoperable soil data sets via web services. The soil information systems that host and publish the soil data of the data providers are referred to as "nodes" in the federation. These nodes could be national (country) systems, regional systems (e.g. the Latin American Soil Information System, SISLAC) or a soil information system of an (inter)national research organization or NGO that wishes to share its soil data. GLOSIS will connect users with providers through a single access point: the discovery hub. The system will allow data providers

three participation levels to make their SIS part of the GloSIS Federation.

1. Tailored implementation – for data holders with an existing SIS. The data holder must implement the GLOSIS data exchange specifications. 2. Template implementation – for data holders that wish to setup and populate their own SIS. They implement the GLOSIS template node. 3. Support implementation – for data holders lacking the resources, knowledge or desire to set up and maintain a SIS. These data holders can submit the soil data they wish to share to the support node.
* * *

---

## Short Comment (SC2) · 18 Oct 2019

Please refer to the INSII Agenda Paper and the GloSIS Conceptual Document here https://docs.google.com/document/d/e/2PACX-1vQkqMTq2B2UoTdYvnj5PmTMRubzb7QZ43hl4M-owr1EUg2mQuzwZN0l25E2qMoiK–wfwMoLRuRfNwf/pub?embedded=true

---

## Referee Comment (RC2) · Anonymous Referee #2 · 22 Nov 2019

This is a nice paper describing an encrypted solution to a distributed database for soil information. Solution to share soil data are always important, given the sensitivity of some soil data and the reluctance of some institutions in sharing data that will be centralised somewhere.

Minor comments: - I think the election process described by the authors is of difficult implementation, not technically but more socially or institutionally. - I could not find mention on how to ensure a minimum data quality (e.g. standardisation, names encoding, laboratory quality). I understand this may be outside the scope of the paper, but I think some discussion about it could be useful. - Re data quality: would be possible to use his system to "grade" the quality of the data? i.e. use different types of institutional keys for different quality of data

---

## Referee Comment (RC3) · Anonymous Referee #3 · 26 Nov 2019

This paper descibes a new protocol for soil data exchange, based on the very popular blockchain technology. After introducing the technology itself, a specific application for a bottom-up global soil spectra library is presented as an illustration of the potentialities of blockchain for soil data sharing. The paper is well scoped and well written, and in particular makes the effort of introducing this complex technology to the soil scientist reader. It could however take a bit more time to explore some of the specific aspects laid down in section 4. I particular, the immutability is brushed off quickly as a great feature, but there need to be explanations of how one could, for example, implement data versioning in such as scheme (in the case of a transcription error that has been spotted, and needs correction, for example). The paper is written in a short and impactful way, which is good. The structure is good too, altough I'd argue that section 2

and 3 should be merged.

However, some parts of the paper need to either be corrected, amended, or added. The part about data ownership is very short and vague, and implementing a technological solution like blockchain does not preclude from having a reflection about data licensing, in my view: blockchain is a technological tool, but the license data is shared under should be acknowledged as the way the rules of engagement between data sharing parties are laid. I also have an issue with the brush statement in the "Data Governance" subsection, which states that "in practice, control and governance over the data are ceded to teh central node". This is simply untrue, if you consider eg federated data management. And one could argue that when the data governance gets decentralised, there is a risk that no governance is going on at all.

Which brings me to my most important problem with the paper as it stands: it ommits completely all the issues associated with blockchain, it does not show the other flip of the coin, so to speak. For example, while the integrity of an asset can be tracked, there is nothing in blockchain that can verify the original certification - in other words, when an actor signs an asset and puts it into the blockchain, there is no mechanism to check whether that asset is legitimate or not. In this case, one could think of a soil spectra that would be invlaid/noisy/faulty. There is also other issues with the technology, like its excessive energy use, its scalability as the database grows.

Don't get me wrong, I'm not trying to say that blockcahin should not be tried or even proposed, rather that a good paper introducing a new technology like bitcoin has to present the drawbacks too, otherwise it runs the risk of "overselling" that technology. There are hard questions to ask about the use of blockcahin in general, and for soil data in particular: is it not completely overkill? Do we want, as a community, to implement a solution we know has a consequent environmental cost, and one that increases with the number of transactions? Is it scalable enough? Integrating a section showing more discussion is, I think, a requirement in this paper to get away from the technological "buzz".

Lastly, with all due respect, I think there are more established papers as a reference for the use of statistical modelling of spectral data than your 2019b paper.

---

## Author Comment (AC2) · 30 Nov 2019

Thanks for your comments Yusuf.

Sadly, we couldn't open the link that you provided. There might be some extra dashes inserted by the SOIL system. We tried all the combinations but it did not work.

Regarding your comment:

**GloSIS is being built in exactly the opposite manner. GLOSIS is envisioned as a federation of soil information systems, which share interoperable soil data sets via web services. The soil information systems that host and publish the soil data of the data providers are referred to as "nodes" in the federation. These nodes could be national(country) systems, regional systems (e.g. the Latin**

**American Soil Information System, SISLAC) or a soil information system of an (inter)national research organization or NGO that wishes to share its soil data. GLOSIS will connect users with providers through a single access point: the discovery hub...**

We think that providing a system like the one you mention, a federation of nodes what are exposed through a single access point, is a great advancement considering what we have at the moment. That system fits very well when the participants do not form a consortium. In terms of concept and design, it is different to what we propose. In your case, for instance, if a node goes off-line for any reason, their data becomes inaccessible. If a node decides to change their data or remove it, that can be done without leaving traces. Of course, all that could be prevented to some degree, but then the system would be closer to either a centralised database or the solution that we propose.

Another point to consider is the independence of the nodes that you mention, that in reality could be limited. For instance, SISLAC has almost 50,000 profile information from Latin America but SISLAC also depends on FAO-GSP. From our experience in Chile, we know SISLAC approaches different groups to request their data to add it to their system and not to promote the development of national soil information systems, which is logical given the nature of the system. Considering the reality in Latin America and other regions, where developing national SIS is not easy, offering regional nodes is a good solution to help data dissemination but, despite the good intentions, it is a centralised (or top-bottom) solution. We prefer to envision a bottom-up solution.

---

## Author Comment (AC3) · 30 Nov 2019

Thanks for your feedback.

Regarding your comment:

**I think the election process described by the authors is of difficult implementation, not technically but more socially or institutionally.**

We agree. Centralised data sharing is easier. You provide your data and then most things are out of your control. Participation requires involvement. We assume that a group of parties willing to start a consortium wants to participate and that the benefits outweigh the extra time required. The main point to consider is that a centralised solution might work. The problem is when it does not work. The consequences can be

very serious. We believe that if we have the option to implement a solution that helps minimise the impact of a problem, we should do it.

**I could not find mention on how to ensure a minimum data quality (e.g. standard-isation, names encoding, laboratory quality). I understand this may be outside the scope of the paper, but I think some discussion about it could be useful.**

We will add some sections to discuss topics that might be somehow independent of the system in place (decentralised or not). How to initiate the consortium (see the response to Dominique Arrouays), define names encoding, how to standardise data, etc. are steps that no system can replace. The solution that we propose can help to enforce some of those decisions but also tries to promote the involvement of the different parties in a more democratic system.

**Re data quality: would be possible to use his system to "grade" the quality of the data? i.e. use different types of institutional keys for different quality of data**

Although it is technically possible to have multiple institutional keys, it is better to avoid that since the idea of having a keys is to identify a party. Nevertheless, having different quality tiers is completely possible. Any information that the consortium decides to add can be added as metadata of a specific data asset.

---

## Author Comment (AC4) · 30 Nov 2019

Thanks for your feedback.

Regarding your comment:

**It [the paper] could however take a bit more time to explore some of the specific aspects laid down in section 4. I particular, the immutability is brushed off quickly as a great feature, but there need to be explanations of how one could, for example, implement data versioning in such as scheme (in the case of a transcription error that has been spotted, and needs correction, for example)**

We mentioned that very briefly at the beginning of Section 3 but we agree that it is not enough. It is possible to create a new transaction to transfer an asset to the same

owner and make changes during that process. Then both versions are permanently linked. We will add extra details on how that works.

**The structure is good too, altough I'd argue that section 2 and 3 should be merged.**

We will merge both sections.

**The part about data ownership is very short and vague, and implementing a technological solution like blockchain does not preclude from having a reflection about data licensing, in my view: blockchain is a technological tool, but the license data is shared under should be acknowledged as the way the rules of engagement between data sharing parties are laid.**

We completely agree. License is important in any system, decentralised or not. We will expand the data ownership section to talk about the importance of data licensing and what a public ledger can offer in that sense.

**I also have an issue with the brush statement in the "Data Governance" subsection, which states that "in practice, control and governance over the data are ceded to the central node". This is simply untrue, if you consider eg federated data management. And one could argue that when the data governance gets decentralised, there is a risk that no governance is going on at all.**

In that paragraph we are specifically talking about a traditional centralised system, that is why we start the paragraph with "In a centralised network, ...". Blockchain is a type of federated data management system, that is why we compare it with the centralised case.

The logic of not providing decentralised data governance because it might fail is interesting. We prefer to stay positive and promote participation.

**... while the integrity of an asset can be tracked, there is nothing in blockchain that can verify the original certification - in other words, when an actor signs an**

**asset and puts it into the blockchain, there is no mechanism to check whether that asset is legitimate or not.**

We omitted topics that are true for any system. The problem you mention is also true for a centralised solution. Adding to the comments of the other reviewers, we will add a few paragraphs to explicitly clarify that this is a technological tool that does not solve human or technical (lab) problems to avoid "overselling" the system by omission.

**There are hard questions to ask about the use of blockchain in general, and for soil data in particular: is it not completely overkill? Do we want, as a community, to implement a solution we know has a consequent environmental cost, and one that increases with the number of transactions? Is it scalable enough? Integrating a section showing more discussion is, I think, a requirement in this paper to get away from the technological "buzz".**

Regarding the question if using a public ledger is an overkill, we do not think it is. Technological solutions are developed to be used. At the moment we share soil data using email and (poorly formatted) Excel files. We need to make a technological jump at some point. The technical part is simpler than it sounds and soil scientist that are closer to those technologies should help to implement them. Of course, there human and institutional factors to consider, but some soil scientist also work closer to those areas. We can see it as a team effort where we are proposing a solution in the technical part.

Regarding the environmental impact, that is a valid concern. Blockchain is a very diverse technology. You mentioned Bitcoin, but that is only one of many public ledgers available to date. The environmental problem is derived from the consensus algorithm that Bitcoin uses (proof-of-work) which is very energy consuming and not scalable. We omitted highly technical details like using a better consensus algorithm (proof-of-stake) that does not require long calculations and high amounts of energy. We will add a paragraph to Section 3 to help dissipate those concern.

**Lastly, with all due respect, I think there are more established papers as a reference for the use of statistical modelling of spectral data than your 2019b paper.**

That is a review where we referenced many established papers. We will add some of those references to the text.